

# Effects of blood parasite infection and innate immune genetic diversity on mating patterns in a passerine bird breeding in contrasted habitats

Dany Garant[1], Audrey Bourret[1], Clarence Schmitt[2], Audrey Turcotte[1], Fanie Pelletier[1] and Marc Bélisle[1]

[1] Département de Biologie, Université de Sherbrooke, Sherbrooke, Québec, Canada
[2] Institut d'Ecologie et des Sciences de l'Environnement de Paris, Sorbonne Universités, UPMC Univ Paris 06, UPEC, Paris 7, CNRS, INRA, IRD, Paris, France

## ABSTRACT

Genetic diversity at immune genes and levels of parasitism are known to affect patterns of (dis)assortative mating in several species. Heterozygote advantage and/or good genes should shape mate choice originating from pathogen/parasite-driven selection at immune genes. However, the stability of these associations, and whether they vary with environmental conditions, are still rarely documented. In this study, we describe mating patterns in a wild population of tree swallows (*Tachycineta bicolor*) over 4 years and assess the effects of haemosporidian parasite infection and immune genetic diversity at β-defensin genes on those patterns within two habitats of contrasting environmental quality, in southern Québec, Canada. We first show that mating patterns were only very weakly related to individual status of infection by haemosporidian parasites. However, we found a difference between habitats in mating patterns related to infection status, which was likely due to a non-random distribution of individuals, as non-infected mating pairs were more frequent in lower quality habitats. Mating patterns also differed depending on β-defensin heterozygosity at AvBD2, but only for genetic partners outside of the social couple, with heterozygous individuals pairing together. Our study underlines the importance of considering habitat heterogeneity in studies of sexual selection.

# INTRODUCTION

Understanding the factors influencing how individuals select their reproductive partners is central in studies of sexual selection and in evolutionary ecology (*Andersson, 1994*; *Jennions & Petrie, 1997*). This undertaking is however challenging as several factors can affect mating patterns in wild populations. For instance, genetic diversity at immune genes and parasitism are both known to affect patterns of (dis)assortative mating in several species (*Hamilton & Zuk, 1982*; *Ejsmond, Radwan & Wilson, 2014*). Theoretical and empirical studies have suggested that such effects may stem from an heterozygote advantage and/or from good

Corresponding author
Dany Garant,
Dany.Garant@USherbrooke.ca

genes that would shape mate choice originating from pathogen/parasite-driven selection at immune genes (*Penn & Potts, 1999*; *Ejsmond, Radwan & Wilson, 2014*).

Previous studies have shown that genes of the major histocompatibility complex (MHC), which are involved in adaptive immune responses (reviewed in *Bernatchez & Landry, 2003*), can affect mating patterns based on the genetic complementarity or diversity of partners (reviewed in *Kamiya et al., 2014*). Genes involved in innate immune responses, such as β-defensin genes (*Van Dijk, Veldhuizen & Haagsman, 2008*), however, have been less investigated in this context despite their importance in providing the first line of defense during an immune challenge, such as an infection by pathogens or parasites (*Delves et al., 2006*). Previous studies in passerines showed variation at different β-defensin genes (henceforth AvBD), which suggest that they could be targeted by selection (*Hellgren, 2015*; *Schmitt et al., 2017a*). In particular, the strongest evidences of selection were found for locus AvBD2 and AvBD7 in both great tits (*Parus major*) and tree swallows (*Tachycineta bicolor*) (*Hellgren, 2015*; *Schmitt et al., 2017a*).

Haemosporidian parasites are ubiquitous and abundant vector-borne blood parasites that cause a malaria-like disease in birds (*Valkiūnas, 2005*). Infections by these parasites often result in negative effects on body condition, breeding success and survival (*Knowles, Palinauskas & Sheldon, 2010*; *Asghar et al., 2015*). For instance, chronic infection by haemospridian parasites resulted in birds laying fewer eggs and being less successful at rearing healthy offspring than uninfected birds in great reed warblers (*Acrocephalus arundinaceus*) (*Asghar et al., 2015*). Thus, resistance to infection from these parasites could be a driving selective force affecting mating patterns in wild bird populations.

So far, only a few studies have investigated the dependence on environmental conditions of mate choice patterns resulting from genetic diversity or parasitism (*Ingleby, Hunt & Hosken, 2010*). This is surprising given the accumulating evidences over recent years that mate choice and sexual selection will often be context-specific and constrained by different ecological factors (see *Robinson et al., 2012*; reviewed in *Miller & Svensson, 2014*). For example, a recent modelling study by *Kaiser et al. (2017)* showed that ecological and social conditions may affect the strength of sexual selection in socially monogamous bird species.

In this study, we describe mating patterns in a wild population of tree swallows over 4 years, in southern Québec, Canada. To do so, we assess whether mating patterns are related to individual status of infection by haemosporidian parasites and if they differ depending on genetic diversity at two immune β-defensin genes. Importantly, we also evaluate if these factors affect mating patterns differentially depending on habitat quality, as this population breeds under contrasted levels of agricultural intensification. Previous studies on this population showed that birds nesting in low quality habitats, dominated by intensive cultures, have a lower breeding success (*Ghilain & Bélisle, 2008*; *Lessard et al., 2014*). Also, tree swallows display a high rate of extra-pair paternity, as about 50% of nestlings are fathered by an extra-pair male (i.e., not the social male; see *Lessard et al., 2014*). Furthermore, previous studies have shown that factors affecting the number of within pair young often differ from the factors affecting number of extra-pair young in tree swallows (*Lessard et al., 2014*) and in other species (reviewed in *Westneat & Stewart, 2003*). These observations support the hypothesis that different mate choice patterns occur

among social and extra-pair couples. We thus assessed how parasitism and innate immune genetic diversity shape mating patterns for both social and extra-pair couples.

## MATERIAL AND METHODS

### Study system and data collection

This study is part of a long-term research project on tree swallows in southern Québec, Canada, which runs since 2004. The study area covers 10,200 km$^2$ and consists of 400 nest boxes distributed equally among 40 farms over a gradient of agricultural intensification (see *Ghilain & Bélisle, 2008* for details). Habitat types found in our study system were classified as intensive farmlands (low quality; e.g., reduced forest cover, loss of marginal habitats and wetlands, and increased field size, resulting in homogenized landscapes) or non-intensive farmlands (high quality; opposite characteristics) based on the percentage of cash crops, such as corn, soybean and other cereals, observed within a 5-km radius around each farm (see *Ghilain & Bélisle, 2008*; *Schmitt et al., 2017b*). Previous studies in this system reported lower reproductive success in intensive farmlands than in non-intensive areas (*Ghilain & Bélisle, 2008*; *Lessard et al., 2014*). All nest boxes are visited every two days during breeding seasons (early May to mid-July). Breeding females and social males are captured and marked at the nest box during incubation and food provisioning, respectively. Blood samples are collected for subsequent DNA analyses. Data were collected in compliance with the Canadian Council on Animal Care, under the approval of the Université de Sherbrooke Animal Ethics Committee (protocols DG2014-01 and FP2014-01).

### Parasite infection screening

For this study, 906 adult tree swallows (465 females and 441 males; 1,260 samples; Table 1) were sampled between 2012 and 2015 and screened for haemosporidian parasite infection, as detailed in *Turcotte et al. (2018)*. Briefly, the detection of avian malaria from bird blood samples was performed with a nested PCR, which consists of two successive PCR amplifications (*Hellgren, Waldenström & Bensch, 2004*). To determine if the PCR amplification was successful, 5 µl of the second PCR product was migrated on a 2% agarose gel stained with ethidium bromide and visualized under UV light. The presence of an infection was confirmed by the detection of an amplification at ca. 500-bp (478-bp for *Leucocytozoon* and 480-bp for *Plasmodium* and *Haemoproteus* without primers).

### β-defensin genes analysis

Of the individuals screened for parasites infection, 69 females and 81 males captured in 2013 and 2014 (on a subsample of 10 farms) were genotyped at β-defensin loci, as detailed in *Schmitt et al. (2017a)* and *Schmitt et al. (2017c)* (see Table 1).

This population of tree swallows has shown variability for several β-defensin genes for both non-synonymous and synonymous SNPs (*Schmitt et al., 2017a*; *Schmitt et al., 2017c*). We used only non-synonymous SNPs because the alleles are potentially responsible for functionally different peptides. We choose to perform analyses of mating patterns using heterozygous status at AvBD2 and AvBD7 loci as these were related to different components of immunity in previous research: heterozygosity at AvBD2 was associated with the absence
**Table 1** Sample sizes for adult Tree swallows (total number of individuals sampled per year) included in analyses and known infection status by haemosporidian parasites or genetic diversity at two β-defensin genes.

|  | 2012 | 2013 | 2014 | 2015 | Total |
|---|---|---|---|---|---|
| Parasite infection |  |  |  |  |  |
|     Females | 132 | 132 | 177 | 173 | 614 |
|     Males | 138 | 144 | 187 | 177 | 646 |
| AvBD2 |  |  |  |  |  |
|     Females |  | 26 | 46 |  | 72 |
|     Males |  | 33 | 63 |  | 96 |
| AvBD7 |  |  |  |  |  |
|     Females |  | 31 | 49 |  | 80 |
|     Males |  | 37 | 65 |  | 102 |

**Notes.**
Total, total number of individuals (including multiple records of individuals across years).

of eggshell bacteria, and homozygosity at AvBD7 with greater innate immune response (*Schmitt et al., 2017c*). Moreover, among the six available β-defensin genes in tree swallows, AvBD2 and AvBD7 were the only loci showing some evidence of selection, with lowest Tajima's D scores (*Schmitt et al., 2017a*). Finally, heterozygosity at both loci was strongly correlated with total heterozygosity when estimated over all loci considered in previous analyses by *Schmitt et al. (2017a)*; Spearman's rank correlation with total heterozygosity: AvBD2: rho $= 0.61$, $P < 0.001$; AvBD7: rho $= 0.44$, $P < 0.001$).

## Mating patterns analysis

Mating patterns were defined for both social and extra-pair couples (males assigned using microsatellite loci—see *Bourret & Garant (2017)* for details on parentage assignment procedures). In brief, candidate genetic fathers of a given nestling included all males captured on the same farm on both previous and following years and within a 15-km radius of the nestling's nest box (covers males from one to nine farms, mean $= 4.8 \pm 1.9$ farms (see *Lessard et al., 2014* for a justification of this scale). However, it should be noted that most paternities in our study system are attributed to males located on the same farm as the female ($\sim$85% see *Lessard et al., 2014*). Assignment rate was 73.8% for nestlings with DNA samples available (93.5% of all nestlings). The rate of extra-pair paternities was 52.3% overall, and did not vary between habitat (low-quality intensive habitats $= 52.3$%, non-intensive $= 52.4$%, $\chi^2 = 0.0002$, $df = 1$, $P = 0.99$).

Patterns were described from the female perspective with respect to (1) their social male and (2) the extra-pair males who fathered some or all of their offspring (genetic males). We compared observed mating patterns to random expectations based on (i) parasite infection status of each individual and (ii) heterozygosity at either AvBD2 or AvBD7 loci. Males were resampled with replacement, within years, and within a buffer of 15 km for the parasite infection status analysis. Observed numbers were compared to 1,000 random expectations for each mating pair status (social and genetic), and *P*-values were calculated using two-tailed distributions (Fig. S1). We used chi-square tests to check for differences in status (i.e., infected or not, heterozygous or not) between social and genetic males, as

well as between habitat types (low or high quality). All analyses were conducted in R (V 3.3.2, *R Core Team, 2017*).

## RESULTS

Overall, 19.5% of social mating partners were infected by haemosporidian parasites (females: 23.7%, males: 16.3%). Females that were not infected by haemosporidian parasites showed mating patterns in accord with random expectations (all $Ps > 0.17$; Fig. 1A). There was a marginally non-significant tendency for infected females to pair with non-infected social males more often than random expectations (social males: $P = 0.094$; genetic males, $P = 0.92$; Fig. 1B). No differences in infection status by haemosporidian parasites were observed between social and genetic males for both non-infected ($\chi^2 = 0.40$, $df = 1$, $P = 0.53$) and infected ($\chi^2 = 0.10$, $df = 1$, $P = 0.76$) females.

Heterozygotes at AvBD2 and AvBD7 represented 34.5% (females: 30.7%, males: 39.5%) and 11.4% (females: 12.9%, males: 9.5%) of social mating partners, respectively. While homozygous females at AvBD2 showed mating patterns that did not differ from random expectation ($P > 0.41$; Fig. 1C), heterozygote females were more likely to pair with heterozygote genetic males than expected by chance ($P = 0.013$); a pattern not found for social males ($P = 0.15$; Fig. 1D).

The level of heterozygosity at AvBD2 did not differ between social and genetic males, when paired with homozygous females ($\chi^2 = 0$, $df = 1$, $P = 1.00$; Fig. 1C). However, social males showed a lower proportion of heterozygous individuals compared to genetic males, when paired with heterozygous females ($\chi^2 = 6.49$, $df = 1$, $P = 0.011$; Fig. 1D). None of the analyses revealed significant differences in mating patterns related to AvBD7 (all $Ps > 0.12$; Fig. S3).

Mating patterns related to haemosporidian parasite infection status differed between habitats for both social and genetic males (all $Ps < 0.001$; Figs. 2A and 2B). This was likely due to a non-random distribution of individuals with different infection status among habitats, as non-infected mating pairs were more frequent in low quality, more intensive, habitats (Figs. 2A and 2B). We observed no differences between habitats in mating patterns related to AvBD2 (all $Ps > 0.83$; Figs. 2C and 2D) or AvBD7 heterozygosity status (all $Ps > 0.30$; Fig. S3).

## DISCUSSION

Our results showed a weak effect of infection status by haemosporidian parasites on mating patterns. There was only a marginally non-significant tendency for infected females to pair with non-infected social males more often than expected by chance. In tree swallows, both females and social males provide food to offspring during the nestling phase (*McCarty, 2002*). Previous studies however provided conflicting results concerning the importance of paternal care in this species with some finding no benefits on females' reproductive success (*Dunn & Hannon, 1992*) and other showing positive effects (*Whittingham, Dunn & Robertson, 1994*). Our results suggest that the choice of social partners by females, who may themselves be limited in their capacity to raise young due to infection by parasites,
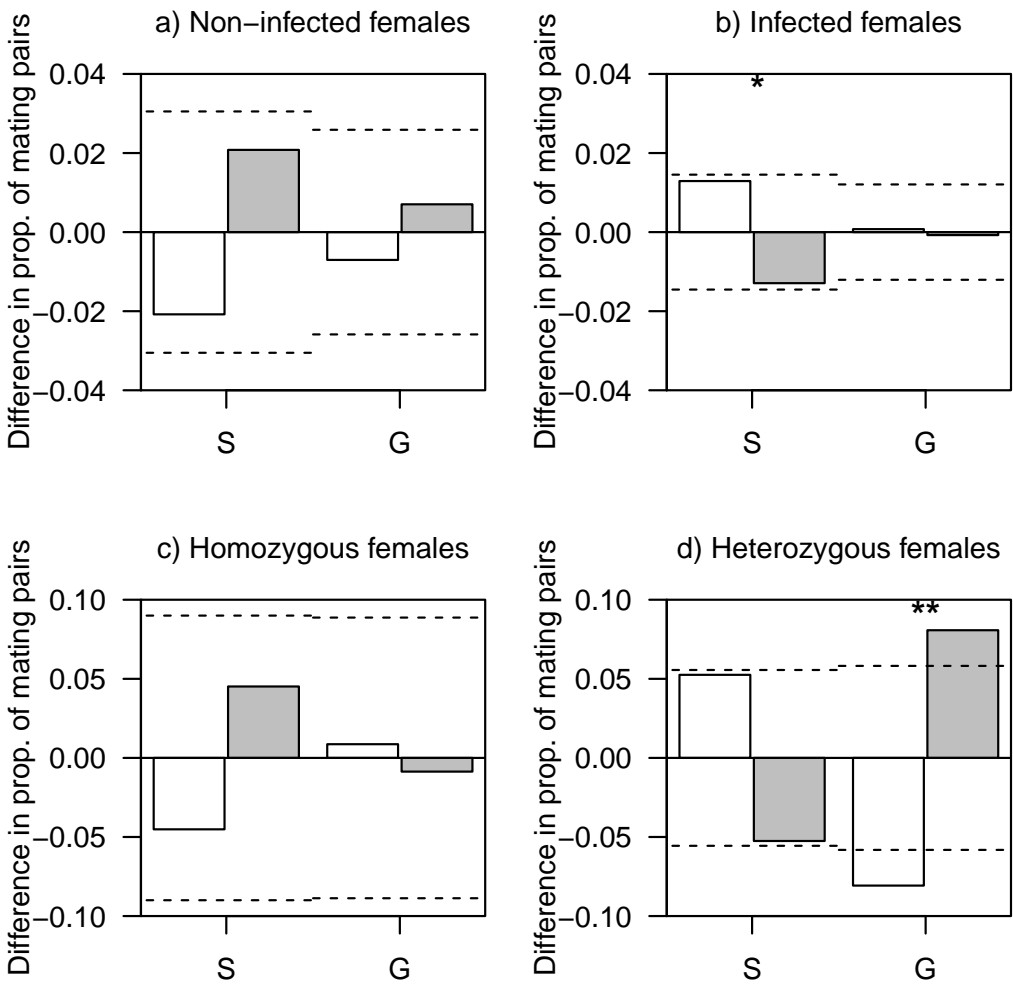

**Figure 1 Difference in proportions of observed and expected mating pairs.** (A) Females non-infected by haemosporidian parasites, (B) infected females, (C) homozygous females at AvBD2 locus and (D) heterozygous females at AvBD2, paired with either social (S) or genetic males (G). White bars represent non-infected (A, B) or homozygote (C, D) males, and grey bars, infected (A, B) or heterozygote (C, D) males. $P$-values ($*P < 0.1$; $**P < 0.05$) are shown. Significance threshold ($P < 0.05$) in each case is represented by a dashed line.

may be somewhat directed toward healthier males that will help with offspring care and competition for nesting sites.

We also showed assortative mating patterns of genetic pairs based on heterozygosity at one of two β-defensin loci. Heterozygous females at AvBD2 were more likely to be mated with heterozygous genetic males at this locus. Several factors could explain this result. For instance, the pattern observed could depend on the female's capacity to assess the genetic diversity of her partners and/or the female's level of choosiness, which could itself be affected by her own heterozygosity (reviewed in *Kempenaers, 2007*). Different signals, such as plumage coloration, are related to male reproductive success in this system (*Van Wijk et al., 2016*) and could thus be used as indicators of heterozygosity. Another possible

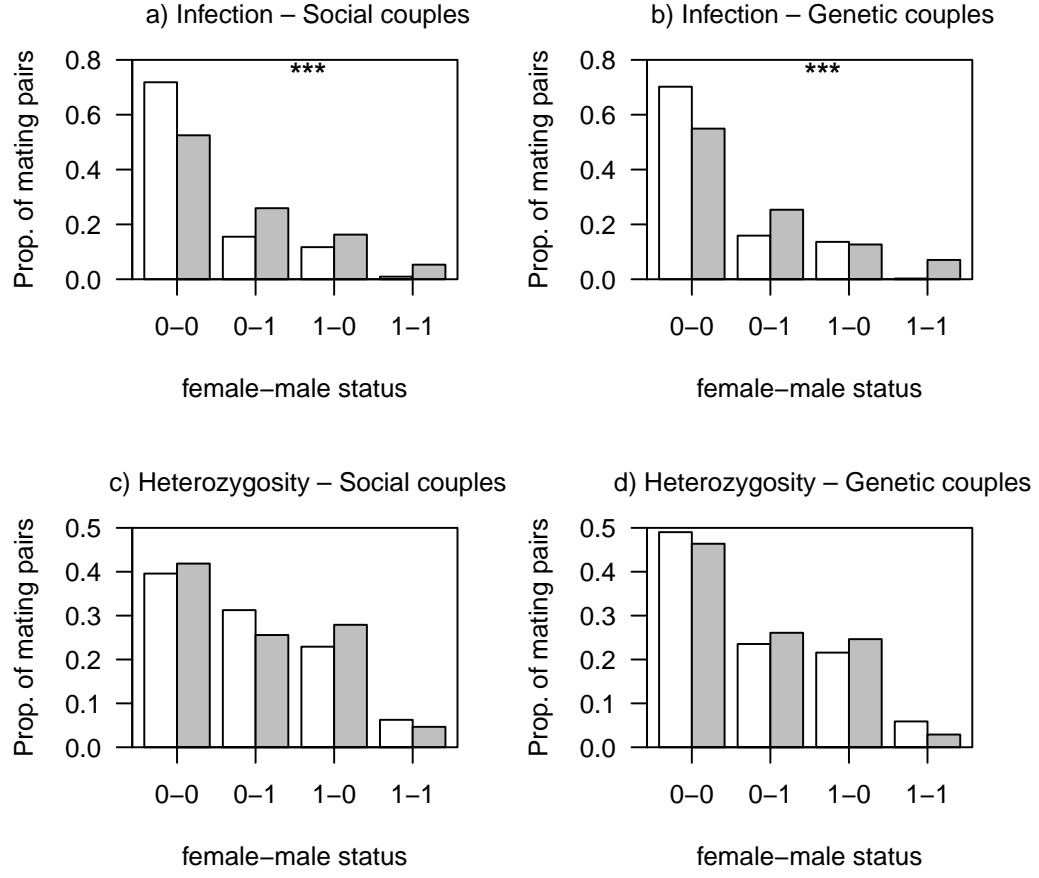

**Figure 2** **Proportions of observed mating pairs between habitats.** White bars: intensive; grey bars: non-intensive as function of infection status by haemosporidian parasites (A, B; non-infected: 0, infected: 1) or heterozygosity at AvBD2 locus (C, D; homozygote: 0, heterozygote: 1), for social couples (A, C) and genetic couples (B, D). *P*-values (***$P < 0.001$) are also shown.

explanation for our result is that patterns of assortative mating are also determined by mutual mate choice, where not only females, but also males, would display preferences for heterozygous partners (*García-Navas, Ortego & Sanz, 2009*; *Kempenaers, 2007*). Our result is thus partly in line with theory suggesting that heterozygosity should be favoured by mate choice (*Brown, 1997*) and previous evidences that females increase heterozygosity of their offspring via extra-pair matings (*Foerster et al., 2003*), including in tree swallows (*Stapleton et al., 2007*). A previous study conducted in this system provided some evidence that heterozygosity at AvBD2 was marginally, negatively associated with the presence of eggshell heterotrophic bacteria present in nests. Additional analyses, however, revealed no evidence for an effect of genetic diversity at this locus on fledging success of females (*Schmitt et al., 2017c*). Further research is needed to conclude on the underlying processes modulating the trend we observed.

We found some evidence that environmental conditions associated with agricultural practices affected the overall distribution of mating patterns. Mating patterns related to

parasite infection status differed between habitats for both social and genetic males, which can be explained by a non-random distribution of individuals, with infected individuals found in greater proportion in non-intensively cultivated habitats. Indeed, additional simulation analyses conducted within each habitat, with smaller sample sizes but allowing to control for differences among habitats in distribution of individuals, revealed that the differences in mating patterns were not significant at the within-habitat scale (all *P*-values > 0.05; see Supplemental Information).

While mate choice patterns are occurring prior to the sampling for infection, most of the infections detected are probably chronic (*Turcotte et al., 2018*). Hence, measuring infection status after mate choice has occurred likely gives a good picture of this aspect at the time mating choices were made. It is probable, however, that some of the infections occurred on the breeding grounds, as we previously showed that local transmission is present in this study system (*Turcotte et al., 2018*). This local transmission could also drive to some extent the non-random distribution of infected individuals we reported here. A previous study in this system by *Turcotte et al. (2018)* showed that non-intensive farmlands were associated with higher *Leucocytozoon* parasites prevalence. These types of land covers correspond to more heterogeneous agricultural landscapes with lower pesticide use (*Ghilain & Bélisle, 2008*), which thus tend to support greater vector and host abundances (see *Bonneaud et al., 2009* for example). Our results are thus partly in line with previous studies showing that mate choice in birds may depend on environmental contexts (*O'Brien & Dawson, 2007*; *Robinson et al., 2012*).

## CONCLUSION

We showed that mating patterns were different depending on heterozygosity at an immune β-defensin locus, but that they only weakly differed according to infection status by haemosporidian parasites. Furthermore, mating patterns related to parasites infection status were different between habitats of contrasted qualities, indicating a possible effect of the environment in driving sexual selection mechanisms. Given that human-driven habitat changes fundamentally modify ecosystems across the globe and can modulate wildlife mating patterns, integrating them into sexual selection studies will provide critical insights on their potential consequences on wild populations.

## ACKNOWLEDGEMENTS

We thank two anonymous reviewers for helpful comments on a previous version of the manuscript. We thank all graduate students and field and laboratory assistants who have contributed to the collection of data and laboratory analysis. We also thank the 40 farmers who provide access to their land each year.

### Funding

This project was supported by grants from the Natural Sciences and Engineering Research Council of Canada (NSERC), the Canada Research Chair program and the Fonds de Recherche du Québec—Nature et Technologies. The funders had no role in study design, data collection and analysis, decision to publish, or preparation of the manuscript.

### Grant Disclosures

The following grant information was disclosed by the authors:
Natural Sciences and Engineering Research Council of Canada (NSERC).
Canada Research Chair program and the Fonds de Recherche du Québec—Nature et Technologies.

### Competing Interests

Dany Garant is an Academic Editor for PeerJ.

### Author Contributions

- Dany Garant, Fanie Pelletier and Marc Bélisle conceived and designed the experiments, contributed reagents/materials/analysis tools, authored or reviewed drafts of the paper, approved the final draft, secured funding.
- Audrey Bourret analyzed the data, prepared figures and/or tables, authored or reviewed drafts of the paper, approved the final draft.
- Clarence Schmitt and Audrey Turcotte performed the experiments, analyzed the data, authored or reviewed drafts of the paper, approved the final draft.

### Field Study Permissions

The following information was supplied relating to field study approvals (i.e., approving body and any reference numbers):

Data were collected in compliance with the Canadian Council on Animal Care, under the approval of the Université de Sherbrooke Animal Ethics Committee (protocols DG2014-01 and FP2014-01).

### Data Availability

Raw data are provided in the Supplemental File.

### Supplemental Information

Supplemental information for this article can be found online at http://dx.doi.org/10.7717/peerj.6004#supplemental-information.

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
