# Peer review of "Effects of blood parasite infection and innate immune genetic diversity on mating patterns in a passerine bird breeding in contrasted habitats"

_PeerJ, doi:10.7717/peerj.6004_

## Round 0.1 · original submission · Minor Revisions

Both the reviewers and I enjoyed your manuscript and agree that it represents a considerable amount of field and lab work as well as a timely contribution to the literature. As you can see in their comments, the reviewers would like to see your hypotheses and predictions better outlined in the Introduction, greater elaboration of your methods, and revised figures that better illustrate the data.

Reviewer 1 ·

Basic reporting

The manuscript is coherent and well written, although see my specific comments below regarding the working hypothesis and predictions, which are not clearly stated in the introduction. The chosen figures are not appropriately labeled, and I made some suggestions to the authors on how to improve their accessibility to readers with minor changes.

Experimental design

The research question was well defined, and the authors did an excellent work in laying out their goal for this study. The experimental design and methods used in this study were chosen well to answer the questions of interest and the materials and methods section is sufficiently detailed to allow its replication.

Validity of the findings

Most of my comments to the authors relate to their findings and the interpretation of their results. I hope the authors will find my comments useful and will use my suggestions to improve the integrity of their manuscript.

Additional comments

This manuscript details an experiment aimed at understanding mating choice in Tree swallows based on three measures of heterogeneity: (1) infection by parasites, (2) β-defensin genetic diversity, and (3) habitat type. The authors differentiated between social and extra-pair male partners and monitored couples during a period of four years.

Specific comments

Social versus extra-pair male partners:
First, the authors need to state their hypothesis and the predictions stemming from it more clearly. Their goal was to assess whether there is evidence for assortative/disassortative mating in Tree swallows based on three different heterogeneity features. The authors explained very clearly how each of these three features may play a role in mate choice, but they have not adequately addressed the distinction between social and extra-pair male partners. The authors should detail their hypothesis on the different selection pressure imposed on the females when choosing a social male partner versus an extra-pair male partner; and provide predictions on how females should choose each of the partner types.

I think that if the authors were to state their hypothesis and predictions more clearly, it would have reflected also on their statistical analysis and interpretation of their results. More specifically, if one assumes that the selection pressure imposed on a female is different when choosing a social partner versus an extra-pair one, then one should distinguish between social partners who are also the genetic partners of the females and between social partners who are not the genetic partners of a specific female. I briefly scanned the raw data the authors provided with the manuscript and noticed that the authors have a significant sample size to test this hypothesis. This distinction may alter the results of this study and should be addressed.

Weak effect:
The authors found no significant mating choice based on individual levels of parasite infection, rather they found an indication of such a trend (p < 0.1). To address this finding as a “weak effect” the authors should provide a power analysis to investigate whether the lack of statistically significant differences is due to small effect size/sample size, and they should report their type II error in this context.

Habitat quality:
The authors need to explain better their terminology in this context. How did they arrive at the conclusion that intense farmland is a low-quality habitat and non-intensive farmland habitat is a high-quality one? They should detail the measures by which they arrived at this conclusion. This is especially important considering that they report the counterintuitive observation of infected individuals preferably inhabiting the habitat they define as high-quality.

Figures:
Two comments about the figures: (1) It would make the figures much more accessible if the authors were to add titles to each panel to indicate what feature of focus is presented there (AVBD2/7, level of parasite infection, etc.); and (2) The authors should consider showing the difference between observed and expected instead of a column for each measure, and then highlight the 0 level. This will make it easier to detect whether groups are statistically different from 0 or not.

Annotated reviews are not available for download in order to protect the identity of reviewers who chose to remain anonymous.

Reviewer 2 ·

Basic reporting

no comment

Experimental design

no comment

Validity of the findings

no comment

Additional comments

This paper describes the results of a four-year field experiment examining patterns of assortative mating with respect to both parasite infection and heterozygosity at two immune genes. In addition to looking at overall patterns, the authors also explore the possibility that mate choice patterns might differ for within vs. extra-pair partners and across habitats that differ in quality. There is a lot of interest in context dependent mate choice and the mechanisms that might maintain variation in mate choice across environments, so this paper is a timely contribution. Moreover, the paper includes a remarkable amount of field data and this robust sample size (especially for the parasite related questions) clearly represents a huge amount of field and lab work. That being said, I do that that as currently written and analyzed there are some issues that make it hard to determine exactly what, if any, differences exist between habitats and hard to interpret exactly what the patterns, or lack of patterns, that the authors describe here mean in a larger context. One of the main conclusions, that there is a ‘difference between habitats in mating patterns related to infection status’ is not at all clear to me. I’ve made several comments below on this point. I found the assortative mating with respect to AvBD result more convincing, though I still had a few questions about its interpretation. Finally, I think there are a number of places where the results, methods, and analyses are somewhat confusing or not fully explained. I’ve detailed these and some more minor comments below.

Specific Comments
• One of the main claims of the paper is that there is a ‘difference between habitats in mating patterns related to infection status’. The authors rightly point out that this is due to a higher number of infected birds in non-intensive habitats. If I’m understanding correctly, the claim that mating patterns differ is based on the fact that the proportion of pairs that are 0-0, 0-1, 1-0, and 1-1 (1 infected, 0 not) is different between the habitats. While the percentage in each category clearly does differ between the two habitats, this seems to be an inescapable product of baseline infection rates and not any clear evidence for different patterns of assortative/dis-assortative mating (which is how it seems to be interpreted). The same pattern would presumably be found for any 1/0 trait that differed in base frequency by habitat whether or not it played any role at all in mate choice. That isn’t to say that infection—and the differences in infection by habitat—aren’t interesting, but I don’t think that the statistical tests currently run are really testing for habitat differences in mating patterns. It would seem that a more fair comparison would be to test for assortative mating within each habitat type (compared to random expectations within that habitat) and ask whether those patterns differ rather than to compare the raw ratios of pair types across the habitats.
• I realize that this study builds on a lot of previous work, but I think that the methods are really not sufficient in several places. Citations to the previous papers are fine, but at least a brief summary of the relevant info would be useful to include here so that the reader does not have to look up a variety of separate references to understand what was done. I’ve noted a few places in particular in the comments where I thought more detail was needed.
• Line 131: What is assignment rate? What is % EPP and does that vary by site/habitat?
• Lines 136-138: What does this mean in terms of the number of farms that were included? Do some farms end up including only the 40 boxes at that farm while others include multiple neighboring farms? Given that infection status is habitat specific, these choices seem very important because it could really alter the ‘random’ expectation for mating patterns. Are the results the same if you randomize only within a farm? How do the number of possible mating choices differ between the farms and what effect might that have on these tests?
• Throughout the text, the authors argue that mating choices might differ with respect to infection status. However, there is no discussion of the fact that the mate choice patterns (putative response) occurred temporally prior to the sampling for infection (putative causal driver of response). Of course if these are chronic infections then measuring them after mate choice has occurred might still give a good picture of what infection status was at the time choices were made, but this isn’t necessarily true if infection is picked up on the breeding grounds (and perhaps this is the case given the habitat dependent associations). This isn’t a problem for the heterozygosity comparisons since those are stable, but the limitation and potential for different interpretations should be discussed.
• Lines 185-190: It isn’t clear to me why the preference would only be exhibited in heterozygous females when presumably (under this model) all females would benefit from a heterozygous mate. Later in the paragraph male preference is mentioned briefly and I suppose this is one mechanism (mutual mate choice) that could produce the pattern here, but the idea is not really fleshed out.
• Figure 1: I found these figures very difficult to interpret and I wonder if there is a better way to visualize these results (even a table might be clearer). It wasn’t clear from the caption why the bars were different heights and that you need to add panels a+b and c+d to get the full sample of pairings. Visualizing the spread of the randomized distribution with respect to the observed point would possibly be more helpful (something similar to figure S1).

Minor Comments
• Line 61: ‘results’ instead of ‘result’ (or change to infections)
• Lines 94: It would be helpful to have a short description of what makes these ‘high’ vs. ‘low’ quality here without having to refer to previous work.
• Table 1: I’m not sure this table adds much. I’m also a bit confused about the ‘total’ column since it doesn’t seem to match the numbers in the methods/results section. Perhaps this one is adding multiple records of individuals across years and the methods reports unique individuals?

---

## Round 0.2 · accepted · Accept

The reviewers and I agree that your revised version has satisfied all concerns and will make an excellent contribution to the literature.

# Reviewer 2 ·

Basic reporting

no comment

Experimental design

no comment

Validity of the findings

no comment

Additional comments

I am happy with the changes that the author's have made in response to my earlier comments.